# Operando real-space imaging of a structural phase transformation in the high-voltage electrode $Li_xNi_{0.5}Mn_{1.5}O_4$

Yifei Sun [1], Sunny Hy[2], Nelson Hua[3,4], James Wingert[3], Ross Harder[5], Ying Shirley Meng [2,6], Oleg Shpyrko[3] & Andrej Singer [1] ✉

Discontinuous solid-solid phase transformations play a pivotal role in determining the properties of rechargeable battery electrodes. By leveraging operando Bragg Coherent Diffractive Imaging (BCDI), we investigate the discontinuous phase transformation in $Li_xNi_{0.5}Mn_{1.5}O_4$ within an operational Li metal coin cell. Throughout Li-intercalation, we directly observe the nucleation and growth of the Li-rich phase within the initially charged Li-poor phase in a 500 nm particle. Supported by the microelasticity model, the operando imaging unveils an evolution from a curved coherent to a planar semi-coherent interface driven by dislocation dynamics. Our data indicates negligible kinetic limitations from interface propagation impacting the transformation kinetics, even at a discharge rate of C/2 (80 mA/g). This study highlights BCDI's capability to decode complex operando diffraction data, offering exciting opportunities to study nanoscale phase transformations with various stimuli.

Over a century ago, Gibbs classified phase transformations into two fundamentally different types based on variations in the order parameter. Continuous transformations exhibit variations "small in degree but may be great in its extent in space"[1]. The initial phase is unstable to infinitesimal fluctuations, resulting in a continuous change of the order parameter across large regions, which can be monitored by measuring macroscopic properties. In contrast, discontinuous transformations display variations "initially smaller in extent but great in degree"[1]. An energy barrier stabilizes the system against infinitesimally small fluctuations until nucleation occurs at a localized region, causing a disruptive change in the order parameter. Capturing intermediate stages of a discontinuous phase transformation is significantly more challenging, as it requires time- and spatially-resolved measurements to observe the nascent nucleus (often only a few nanometers large) and the subsequent interface propagation during growth. Yet, discontinuous phase transformations are of utmost importance in materials science because properties can be tuned by balancing nucleation and growth to achieve the desired microstructure[2,3].

X-ray powder diffraction proves to be a powerful tool for examining phase transformations with sufficient time resolution. During a phase transformation, the diffraction condition differs between the coexisting phases, and each crystalline phase produces a distinct Debye-Scherrer ring[4–6]. Recent advances in high-brilliance synchrotron sources have enabled in-situ and operando measurements to capture diffraction signals from individual sub-micron particles[7]. In particular, Bragg Coherent Diffractive Imaging (BCDI) measures 3D strain distribution and buried defects within individual nanoparticles by inverting coherent diffraction data to real space structural information via an iterative phase retrieval algorithm[8–12]. In principle, BCDI offers an unparalleled opportunity to image nucleation and growth in individual particles under operating conditions, all without the need for specialized sampling environments. Nevertheless, the challenge of inverting complex diffraction data comprised of separated diffraction peaks has hindered the realization of real space imaging of discontinuous phase transformations with BCDI[13,14].

[1]Department of Materials Science and Engineering, Cornell University, Ithaca, NY, USA. [2]Department of Nanoengineering, University of California San Diego, La Jolla, CA, USA. [3]Department of Physics, University of California San Diego, La Jolla, CA, USA. [4]PSI Center for Photon Science, Paul Scherrer Institute, Villigen, Switzerland. [5]Advanced Photon Source, Argonne National Laboratory, Argonne, IL, USA. [6]Pritzker School of Molecular Engineering, University of Chicago, Chicago, IL, USA. ✉e-mail: asinger@cornell.edu

Here, we show BCDI's capability to image a discontinuous phase transformation induced by Li-intercalation in a single 500 nm large $Li_xNi_{0.5}Mn_{1.5}O_4$ $(0 < x < 1)$ particle inside an operational Li metal coin cell. Like many other high-rate positive electrode materials, the disordered spinel $Li_xNi_{0.5}Mn_{1.5}O_4$ experiences phase separation from the discontinuous phase transformation during cycling[15–18]. Through the transformation, internal stresses arise at the migrating interface that can impede kinetics and lead to mechanical degradations[19]. Yet $Li_xNi_{0.5}Mn_{1.5}O_4$ shows great potential to outperform its commercial opponents in terms of stability and cost[20,21].

Using a newly developed correlated phase retrieval algorithm[14] to decode the phase-separating diffraction patterns, we directly observe the nucleation and growth of the Li-rich phase inside the initial, fully charged Li-poor phase. Operando imaging reveals the transformation of a curved coherent interface into a planar semi-coherent interface, driven by the introduction of dislocations. Supported by the micro-elasticity model, our data show that the dislocation array at the semi-coherent interface reorients the interface during operation. We see no evidence of kinetic limitation from the interface propagation on the phase transformation, even at a discharge rate of C/2 (80 mA/g, full discharge in 2 h). Our study unveils BCDI's potential as a robust tool for operando insights into discontinuous structural phase transformations in nanoscale systems, whether induced by electrochemical processes, temperature fluctuations, optical light exposure, or electronic excitation.

## Results

Figure 1a shows the experimental setup for the operando measurements (Figs. S1–2). During discharge, Li-ion intercalation induces a structural phase transformation from the Li-poor phase (smaller lattice constant, $d_c$) to the Li-rich phase (larger lattice constant, $\mathbf{d}_d$)[4,18]. The different lattice spacing results in two separate Bragg reflections, $G_{III,c}$ and $\mathbf{G}_{III,d}$. When the two phases coexist inside a single crystal, both Bragg reflections are present (Fig. 1b). The scattering intensity around a Bragg reflection is decorated by an interference pattern due to the illumination from spatially coherent x-rays on a single particle[8]. For a particle with coexisting phases, the superposition of the interference patterns captures the spatial distribution and the relative crystallographic registry of both phases, as well as the structure of the interface. To measure the 3D interference patterns, we recorded a series of 2D sections of the Ewald sphere across the reciprocal space by rocking the operando cell (Fig. S3)[22].

Figure 1c–j presents the central slice of the measured operando 3D diffraction data as a function of depth of discharge (DoD, in equilibrium equivalent to $x$ in $Li_xNi_{0.5}Mn_{1.5}O_4$) for the 111 Bragg peak of both phases (see Fig. S4 for the full dataset with 50 measurements). At 0% DoD, corresponding to the fully charged state, only one Bragg peak surrounded by interference fringes appears at the larger momentum transfer, $Q_{III}$, normal to the (111) planes (Fig. 1c). During the initial stages of discharge, spanning from 0% to 35% DoD, the main peak shows variations in its interference fringes, suggesting local structural changes within the nanoparticle due to lithium intercalation (Figs. 1d and S4). From 0% to 15.3% DoD (Fig. S4), a flare appears on the main peak. We interpret this flare as x-ray interference due to changes in the displacement field from Li intercalation, rather than the emergence of a second structural phase. There is a slight decrease of the peak position along $Q_{III}$, indicating a solid solution behavior during the initial introduction of lithium[23]. Starting between 41.5% and 50.5% DoD (Fig. 1e, f), a secondary peak emerges around $Q_{III} = 1.34\,Å^{-1}$, steadily gaining intensity at the cost of diminishing intensity in the primary peak (Fig. 1e–i). This is the hallmark of a discontinuous structural phase transformation, characterized by a substantial variation in the order parameter (namely, the lattice constant). The presence of both diffraction peaks emanating from an individual sub-micrometer crystal indicates phase coexistence.

Notably, the presence of the coexisting peaks in the diffraction data occurs at the voltage plateau in the electrochemical data[24] (Fig. S5), consistent with previous operando x-ray diffraction on $Li_xNi_{0.5}Mn_{1.5}O_4$[4,10].

To interpret the measurements, we use the recently developed correlated phase retrieval algorithm[14] to invert the diffraction data into real-space 3D images. Critical for the success of the algorithm is inverting a series of operando measurements simultaneously, while assuming an approximately static shape of the particle across different discharge states (Fig. S6). We argue that the assumption is true here because the material is stable for hundreds of charge-discharge cycles, suggesting retention of the particle shape during a single discharge. This assumption is also supported by the BCDI data on $Li_xNi_{0.5}Mn_{1.5}O_4$ before and after phase transformation[23]. The algorithm reconstructs the particle shape and the 3D displacement field along the scattering vector $Q_{III}$[10,11,25]. Subsequently, we derive the 3D strain distribution from this displacement field through numerical differentiation along $Q_{III}$[23]. The strain maps are consistent across different independent phase retrieval runs on mutually distinct subsets of the operando data (Fig. S7, S9), and the reconstructed diffraction patterns exhibit good agreement with the measured diffraction data (Fig. S8), affirming the success of the phase retrieval procedure.

Figure 2 illustrates the operando strain evolution within a $Li_xNi_{0.5}Mn_{1.5}O_4$ nanoparticle, as obtained from the phase retrieval on the operando diffraction data (see Figs. S9, 10 for the full dataset of 50 images). Overall, the imaging data portrays the nucleation and growth of the Li-rich phase at the expense of the Li-poor phase through interface advancement (similar behavior is observed in another $Li_xNi_{0.5}Mn_{1.5}O_4$ particle, see Figs. S11, 12). At the onset of discharge, the nanoparticle consists of an almost homogeneous Li-poor phase (negative strain, depicted in purple) (Fig. 2a–c). As the electrochemical lithiation proceeds, an inclusion of the Li-rich phase (positive strain, depicted in yellow) nucleates at the bottom right corner of the nanoparticle (Fig. 2d). At this stage, the imaging suggests the presence of multiple nucleation sites (Fig. S10); however, as the growth proceeds, only a single inclusion prevails (Fig. 2e). The merging is likely driven by surface tension, akin to Ostwald ripening or coarsening[26]. Throughout the subsequent lithiation, the Li-rich phase grows at the expense of the Li-poor phase via interface propagation (Fig. 2e–n). We approximate the interface propagation velocity to be 0.17 nm/s, notably slower than the expected Li-ions diffusion inside the particle (Fig. S13). Consequently, the Li ions have sufficient time to equilibrate concentration gradients within each phase, likely resulting in a sharp Li-concentration gradient at the interface[27]. This is consistent with our observation that the interface width between the coexisting structures in Fig. 2i, j is less than 100 nm, approaching the spatial resolution of operando BCDI[9,11]. By the end of the process, the nanoparticle is comprised entirely of the Li-rich phase (Fig. 2o).

In addition to visualizing in real time the nucleation and growth of a secondary phase, operando BCDI provides insights into the morphological evolution of the interface between the coexisting phases. Our data shows an initially curved interface (Fig. 2e, f) that subsequently transforms into a planar configuration (Figs. 2j, k and S10). This morphological change is likely associated with the microstructure dynamics at the interface. Heterointerfaces between two distinct crystalline structural phases are typically classified into three types: coherent interfaces maintaining complete continuity of the lattice (Fig. 3a), semi-coherent interfaces with piecewise continuity separated by misfit dislocations (Fig. 3d), and incoherent interfaces with no registry between the two phases[28]. In our single-particle diffraction data, the coexisting phases generate diffraction peaks in proximity (less than $0.01\,Å^{-1}$ in $Q_\perp$ in Fig. 1e–h). Therefore, the two crystalline phases are closely aligned, ruling out a fully incoherent interface, which typically occurs when the crystal planes are misaligned by more than 15 degrees[29].

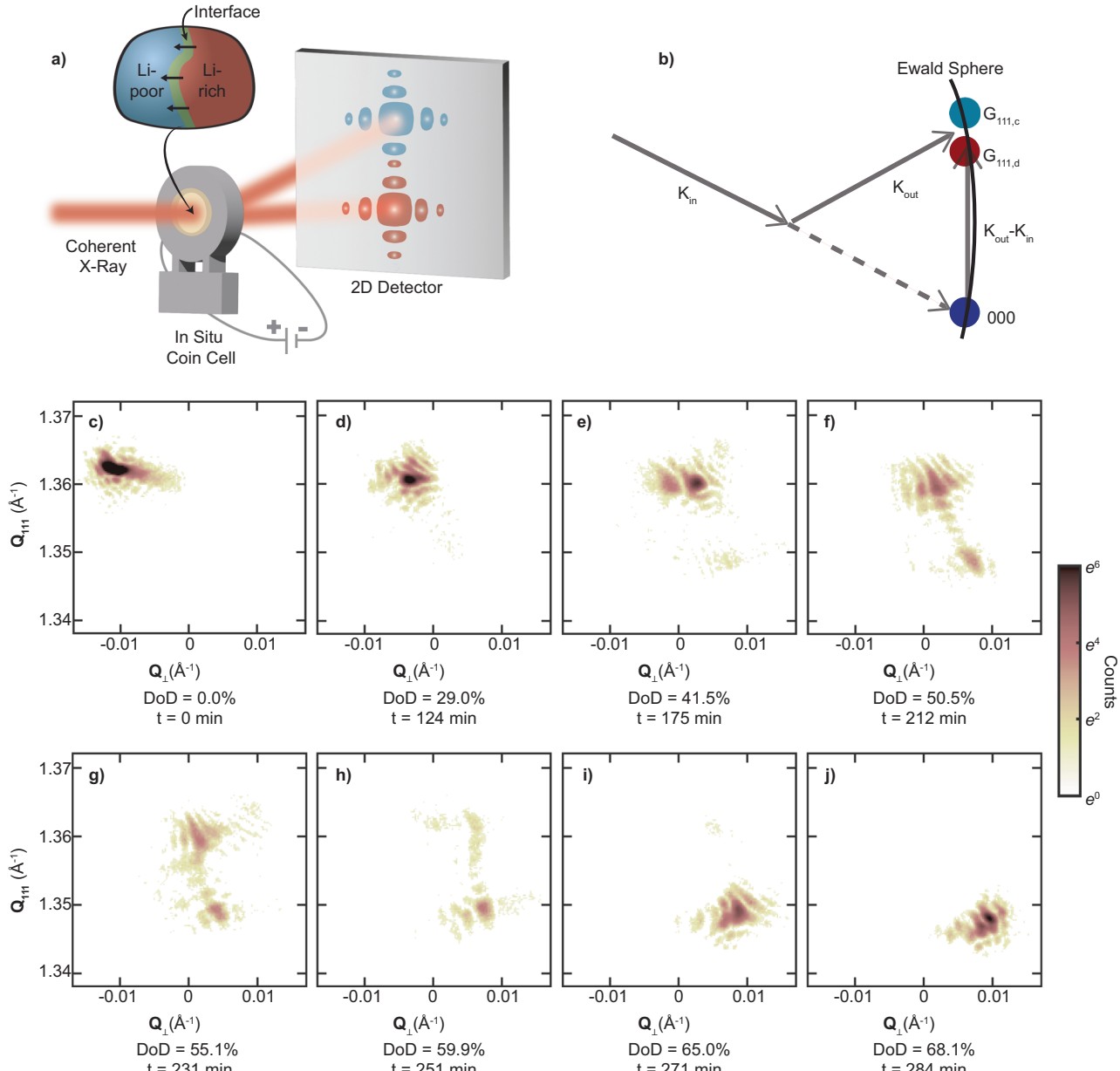

**Fig. 1 | Operando Bragg Coherent Diffraction of a single $Li_xNi_{0.5}Mn_{1.5}O_4$ nanoparticle during the discontinuous solid-solid phase transformation induced by electrochemical Li-insertion. a** Experimental setup showing the operando coin cell, illuminated by coherent X-rays at 9 keV with a focus size of 800 nm, and diffraction around the 111 Bragg peak recorded on an area detector. During discharge, the $Li_xNi_{0.5}Mn_{1.5}O_4$ particle undergoes a discontinuous phase transformation via phase coexistence of the Li-poor phase (in blue) and Li-rich phase (in red), separated by the interface (in green). **b** Schematic illustrating the Ewald sphere construction and mapping out the interference profile surrounding the reciprocal

lattice points $G_{111,d}$ (Li-rich phase) and $G_{111,c}$ (Li-poor phase). $K_{in}$ is the incident X-ray, and $K_{out}$ is the diffracted X-ray. The area detector records a segment of the Ewald sphere. As it intersects both reciprocal space vectors, the detector image shows a split peak. **c–j** Cross-sections of the 3D diffraction pattern for the same $Li_xNi_{0.5}Mn_{1.5}O_4$ nanoparticle at various depths of discharge (DoD) and time (t). The depth of discharge is defined as the fraction of the capacity that is currently removed from its full capacity. The vertical axis points along the scattering vector, $\mathbf{Q}_{111}$, and the horizontal axis ($\mathbf{Q}_\perp$) points perpendicular to $\mathbf{Q}_{111}$.

To distinguish between the coherent and semi-coherent hetero-interface, we adopt the microelasticity model within the framework of continuum mechanics, which was adapted to study phase transformation in $Li_xFePO_4$ ($0 < x < 1$), another technologically important phase separating positive electrode material[30,31]. For an inclusion of a secondary phase in a matrix, the strain energy is related to the function of direction, $B(\mathbf{n})$, which incorporates elastic properties of the system and crystallography of the phase transformation[32]. Minimization of the total strain energy requires aligning the interfaces of the inclusion with minima in $B(\mathbf{n})$. Here, we calculate $B(\mathbf{n})$ for an inclusion of the Li-rich

phase inside the Li-poor phase (using Einstein notation for summation)

$$B(\mathbf{n}) = \lambda_{ijkl}\varepsilon_{ij}^0\varepsilon_{kl}^0 - n_i\sigma_{ij}^0\Omega_{jl}(\mathbf{n})\sigma_{lm}^0\mathbf{n}_m \tag{1}$$

where $\mathbf{n}$ is the interface normal, $\varepsilon_{ij}^0$ is the strain tensor, $\sigma_{ij}^0$ is the stress tensor, $\Omega_{ij}$ is related to the elastic Green's tensor and defined as $\Omega_{ij}^{-1} = \lambda_{iklj}n_kn_l$, and $\lambda_{iklj}$ is the elastic stiffness tensor. Since the Li-poor phase transforms into the Li-rich phase while maintaining its cubic (spinel) symmetry[18], the transformation strain is almost isotropic and around 0.9% based on the difference in lattice parameters. The

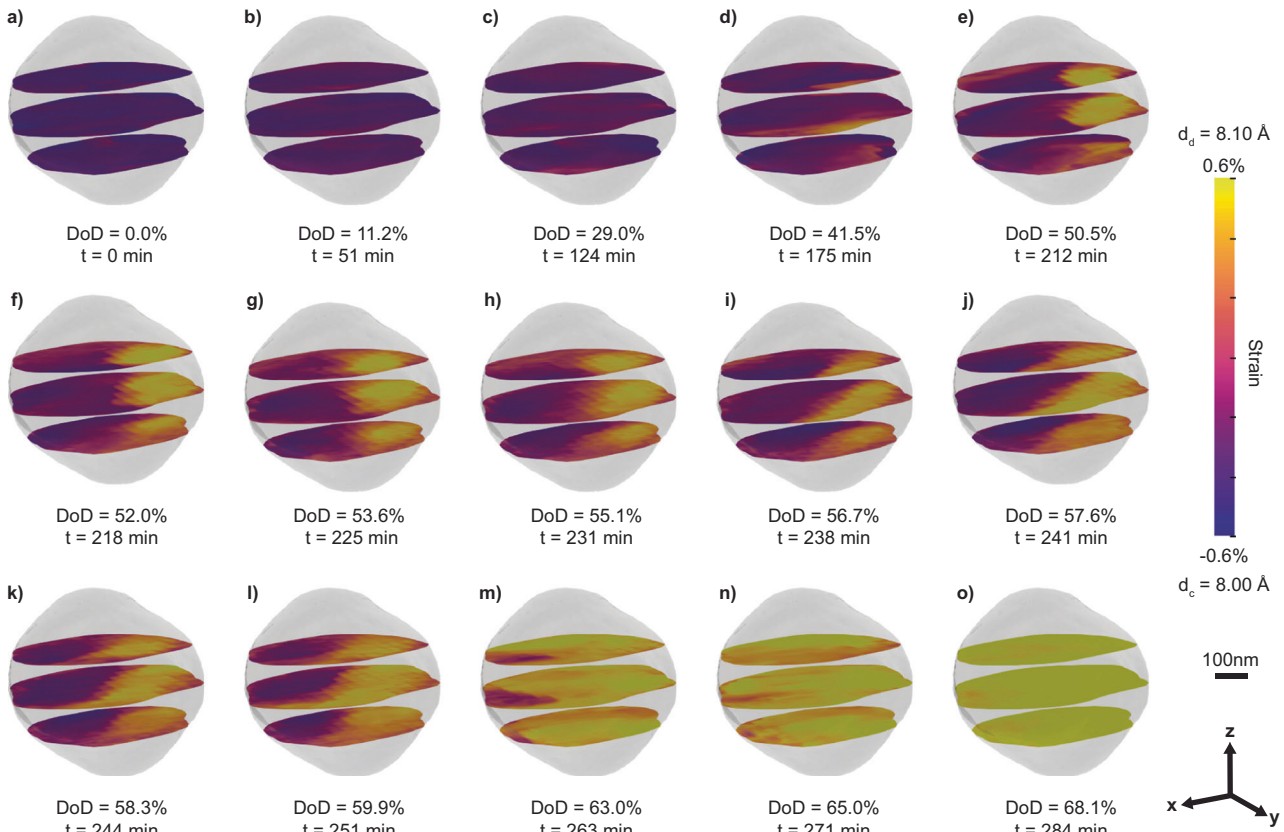

**Fig. 2 | Operando visualization of the coexisting phases during the structural phase transformation under lithium intercalation. a–o** The 3D strain field inside a single $Li_xNi_{0.5}Mn_{1.5}O_4$ nanoparticle during the two-phase reaction induced by Li intercalation. The strain maps are extracted by inverting coherent X-ray diffraction data shown in Figs. 1c–j and S4. The semi-transparent isosurface portrays the particle shape, while the colored slices display the strain distribution, $\varepsilon_{111}(\boldsymbol{r})$, on three chosen planes. We show the strain field, $\varepsilon_{111}(\boldsymbol{r})$, as the local lattice constant compared to the average lattice constant, $d_a$, of the (111) planes between the fully

charged, Li-poor phase ($d_c = 8.00\text{Å}$) and the partially discharged Li-rich phase ($d_d = 8.10\text{A}$), where $d_a = (d_c + d_d)/2 = 8.05\text{ A}$ and $\varepsilon_{111}(r) = d(r)/d_a - 1$. At 0% DoD in **a**), the particle has a uniform negative strain shown in dark purple, corresponding to the Li-poor phase with a small lattice constant, $d_c$. At the end of the two-phase reaction around 68.1% DoD in (**o**), the particle presents a uniform positive strain shown in light yellow, corresponding to the Li-rich phase with a large lattice constant $d_d$. $Q_{111}$ points along z.

calculated $B(\mathbf{n})$ from Eq. (1) reveals a minimum direction along the family of <100> directions (Fig. 3b, c). Notably, the maximum direction of $B(\mathbf{n})$, <111>, differs from the minimum direction, <100>, by a mere 8%. Given this weak dependence of direction on $B(\mathbf{n})$, the modeling suggests an overall spherical shape of the interface, consistent with our observation in Fig. 2e–h.

As the structural transformation advances and the interfacial area expands, significant coherency strain arises[28,30]. To alleviate this strain, the interface can introduce misfit dislocations and form a semi-coherent interface (Fig. 3d). In this configuration, the interface comprises regions that maintain lattice continuity and is interspersed with dislocation cores that release interfacial strain and disrupt this continuity[28]. These misfit dislocations introduce an anisotropy to the transformation strain, consequently altering the orientations of the low-energy phase boundaries[30,31]. The transformation from coherent to semi-coherent interface here is reminiscent of the coherency loss in metals[33]. For the spinel structure, the slip directions and planes are <110> and {111}[34]. We assume that the dislocations possess a Burgers vector along [110] and lie within the ($\bar{1}11$) plane (Fig. S14). As a result, these dislocations lead to a loss of coherency along [110]. Setting $\varepsilon_{110} = 0$, the calculation shows the minimum $B(\mathbf{n})$ along [$\bar{1}10$] (Fig. 3e), and the difference between the lowest and highest $B(\mathbf{n})$ directions has increased to 197% (Fig. 3f). This strong anisotropy in $B(\mathbf{n})$ leads to a preferential interface orientation along [$\bar{1}10$], and correspondingly, the microelasticity

model predicts a transition of the interface geometry from a curved to a planar configuration, as we observe in Fig. 2h, i.

To validate the transition from a coherent to a semi-coherent interface, we examine the presence of dislocations at the interface after it transforms into the planar geometry. Dislocations manifest as singularities in the displacement field[10,11,25]. At the fully charged state (DoD = 1.2%), the particle exhibits uniform negative strain, and the displacement field is continuous without singularities (Fig. 4b). As the discharge progresses, the Li-rich phase with positive strain forms, leading to changes in the displacement field (Fig. 4c). At the interface, the continuity of the displacement field remains uninterrupted, showing no indications of structural defects such as dislocations. Yet, for the subsequent discharge, when the planar interface forms, the displacement map in Fig. 4d reveals an array of misfit dislocations. The presence of these dislocations corroborates our earlier hypothesis of a semi-coherent interface for strain relaxation in the microelasticity model. In this diffraction geometry, the phase discontinuity we measure is the projection of the Burgers vector along [111]. Our data is consistent with a Burgers vector along [110] generating a phase discontinuity of 5.1 radians (Fig. S15). Given a 0.9% lattice mismatch and the slip for each dislocation along the interface, we approximate one dislocation per 52 nm, which is of the same order of magnitude as the observed dislocation density of about one per 95 nm in Fig. 4d. The microelasticity model predicts a minimum-energy interface with a normal along [$\bar{1}10$] (Fig. 3e, f), consistent with the strain map in Fig. 4d

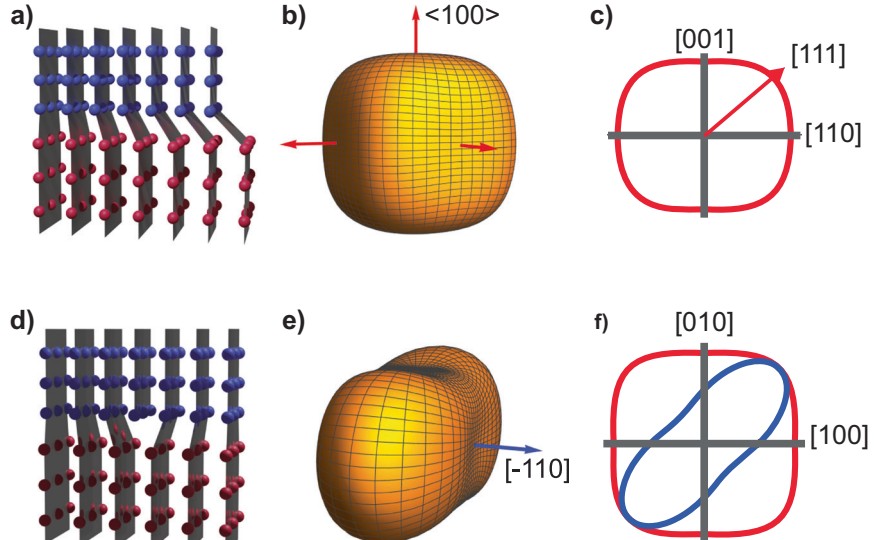

**Fig. 3 | Microelasticity theory for coherent and semi-coherent interface in Li$_x$Ni$_{0.5}$Mn$_{1.5}$O$_4$. a** Illustration showing a coherent interface. Each plane of blue atoms representing the Li-poor phase is connected to a plane of the red atoms representing the Li-rich phase. The lattice distortion gradually intensifies from the left to the right. **b** Isosurface plot of the elastic strain energy as a function of normal direction for a coherent interface. The red arrows indicate the energy minima directions, <100>. **c** 2D parametric plot of the strain energy illustrated in (**b**) in the plane spanning the [001] and [110] directions. The energy minimum, [001], and maximum direction, [111], display similar values. **d** Illustration of a semi-coherent interface with a misfit edge dislocation—one extra half-plane inserted from the top—that relieves the misfit strain in the direction of the Burgers vector perpendicular to the half-plane. **e** Isosurface plot of elastic strain energy for a semi-coherent interface with coherency loss in the [110] direction. The blue arrow indicates the energy minimum direction along [1̄10]. **f** 2D parametric plot in the plane spanning [100] and [010], comparing strain energy between a coherent (red) and a semi-coherent (blue) interface. The semi-coherent interface exhibits significant anisotropy in its strain energy.

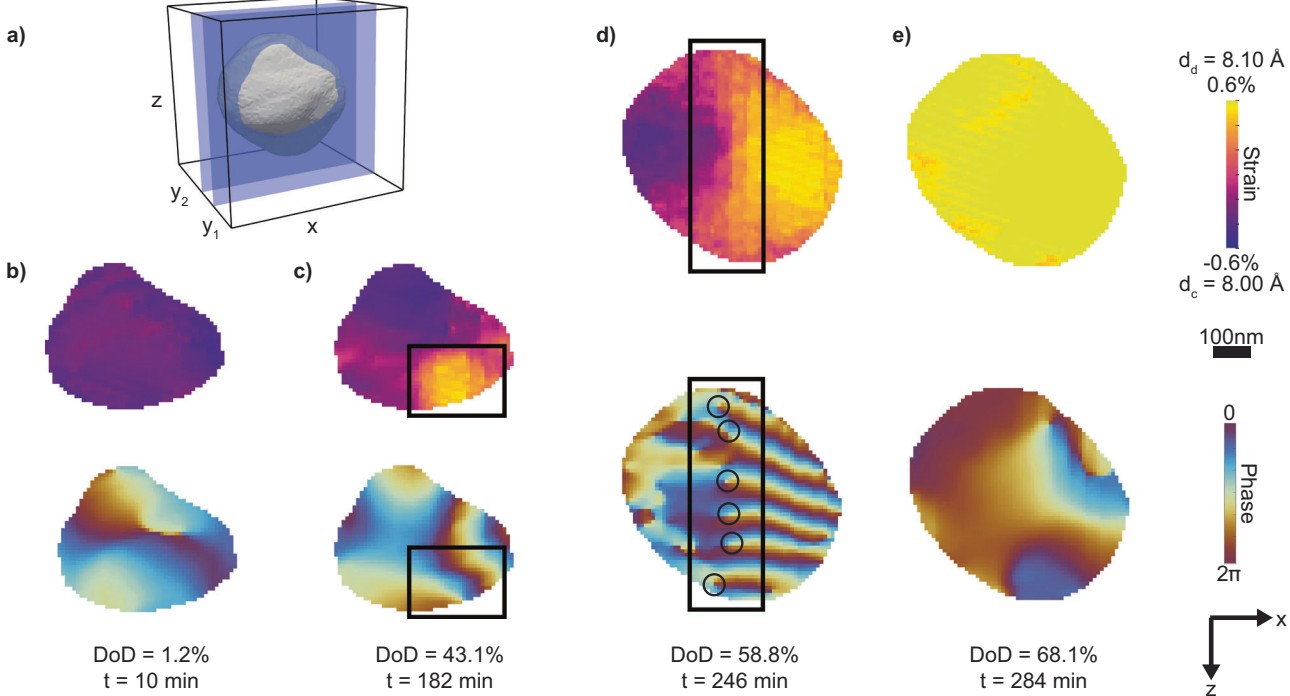

**Fig. 4 | The cross-sections of the reconstructed strain and displacement field in the xz plane at different depths of discharge. a** Three-dimensional representation of the Li$_x$Ni$_{0.5}$Mn$_{1.5}$O$_4$ particle aligned at the same angle as in Fig. 2 but with two slices taken along a different direction. **b–e** Reconstructed strain (top) and lattice displacement (bottom) maps. **b** Prior to the phase transformation, the strain map corresponds to a fully charged Li-poor phase. The false color in the phase maps reflects the displacement from the ideal lattice at each position; a displacement of 2π corresponds to the (111) lattice spacing. **c** At the early stage of the phase transformation, the Li-rich phase (enclosed by the rectangle) nucleates at the lower right corner, leading to an intensified color gradient within the rectangle. Both xz slices in (**b**) and (**c**) are located at y$_1$ in (**a**), 170 nm away from the center of the particle where the discharged phase starts to nucleate. **d** During the phase transformation, the nucleated Li-rich phase forms a semi-coherent interface with the Li-poor phase (outlined by the rectangle), which contains an array of dislocations each circled in the phase map. The dislocations run into the page, and the semi-coherent interface propagates along the negative x-direction. **e** At the end of phase transformation, the strain map shows the Li-rich phase with no observable signs of dislocations in the phase map. The xz slices in (**d**) and (**e**) are located at y$_2$ in (**a**), at the center of the particle. The scattering vector, Q$_{111}$, points along the z-axis.

showing the interface orientation almost normal to the scattering vector $Q_{III}$.

## Discussion

The access to crystal microstructure in our operando data provides insights into the nanomechanics at the propagating interface. The direction of dislocation motion has a significant component perpendicular to the Burgers vector. This motion is reminiscent of dislocation climb, a process requiring diffusion of host species (Ni, Mn, and O). Yet, $Li_xNi_{0.5}Mn_{1.5}O_4$ is reversible for hundreds of cycles[35], so a diffusional flux of host species during every cycle seems implausible. Drawing inspiration from the well-established theory of phase transformations in metals[28], we propose a model involving a glissile motion of interfacial dislocations. In this framework, the interface is generated through an invariant shear deformation and dislocations enter the crystal at the surface along the slip planes created by the shear (Fig. S16)[29]. Thus, the interface can propagate conservatively without the kinetically-limited diffusion of host species[28]. As we see in the imaging of a $Li_xNi_{0.5}Mn_{1.5}O_4$ nanoparticle measured at a five times higher discharge rate, C/2 (80 mA/g), (Fig. S17), phase separation is still evident in the strain maps despite that the two-phase reaction completes within 40 min (Fig. S18). This suggests a transformation mechanism that is not limited by lithium diffusion but rather by the externally applied current that determines the discharge rate. Our results are consistent with recent phase-field modeling, which suggests that interfacial coherency loss can substantially improve reaction kinetics for high-rate positive electrode materials when the two-phase reaction is unavoidable[31]. When the phase transformation completes, the interfacial dislocations move along with the interface and exit the nanoparticle as Fig. 4e shows no dislocations.

Various models have been proposed to explain intercalation-driven phase transformations in rechargeable battery electrodes. Generally, ion transport inside the electrode and ion insertion kinetics across the electrode/electrolyte interface determine the phase transformation mechanism[27]. At fast (dis)charge rates (typically over 1C), the transformation kinetics is limited by ion diffusion, and the shrinking core model was proposed to describe the phase separation during the transformation in $Li_xFePO_4$[36–38]. At slower (dis)charge rates (typically under 1C), the transformation kinetics is limited by the charge transfer across the surface of the electrode. Under this regime, models like the domino cascade can explain the transformation of anisotropic ionic diffusion materials[39,40]. For materials such as $Li_xNi_{0.5}Mn_{1.5}O_4$ with a cubic crystal structure, ionic diffusion is typically isotropic. Nevertheless, our operando imaging shows that the reaction nucleates from a single localized point and grows through interface propagation[41], even though most of the particle's surface is in contact with the liquid electrolyte. At (dis)charge rates C/2 (80 mA/g) and C/10 (18 mA/g) we investigated, the transformation occurs in the charge transfer limited regime, where the ion diffusion within the particle is faster than its transfer from the electrolyte to the particle. In this case, the gradient in the chemical potential is small, and nucleation is rare due to the small driving force, making the reaction nucleation limited. This contrasts with the diffusion-limited regime, where a larger driving force leads to many nucleation sites, but their growth is constrained. During the phase transformation, we directly observe transient misfit dislocations at the interface, which relax the large misfit strain of 0.9% between the coexisting phases and likely prevent extensive cracking and fracture during the phase transformation. The speed of this transformation is controlled by the externally set discharge rate, free from intrinsic kinetic limitations due to the transient coherency loss at the interface.

Reflecting on Gibbs's classification, our research unveils BCDI as a powerful tool to study discontinuous phase transformations operando. Moreover, the recently developed diffraction-limited synchrotron sources will boost BCDI's time resolution to seconds[42], enhancing our ability to observe these transformations at faster rates. Finally, an increased diffraction signal will yield higher spatial resolution and enable the detection of diffraction from potentially present interphases during the phase transformation; at our current resolution, we observed no such phases. Our results, therefore, unlock a feedback loop between stimuli and characterization, which is critical for balancing the nucleation and growth in nanomaterials for optimizing materials' properties.

## Methods

### Sample synthesis and coin cell assembly

$LiNi_{0.5}Mn_{1.5}O_4$ disordered spinel was synthesized using the sol-gel method[43]. The sol solutions were prepared from the stoichiometric mixtures of $Li(CH_3COO) \cdot 2H_2O$ (Fisher, 99% purity), $Ni(CH_3COO)_2 \cdot 4H_2O$ (Fisher, 99% purity), and $Mn(CH_3COO)_2 \cdot 4H_2O$ (Fisher, 99% purity) in distilled water. Next, the solution was added dropwise to a continuously stirred aqueous solution of citric acid. The pH of the mixed solution was adjusted to 6.5 by adding an ammonium hydroxide solution. The solution was then heated at 75 °C overnight; a transparent gel was obtained. The resulting gel precursors were decomposed at 500 °C for 12 h in air and then calcinated at 900 °C for 14 h in air.

For the composite electrode fabrication, the slurry consisting of 80 wt% active materials, 10 wt% acetylene carbon black (MTI, battery grade), and 10 wt% poly(vinylidene fluoride) (PVdF) (Arkema, battery grade) in N-methyl pyrrolidone (NMP) (Sigma-Aldrich, 99.5% purity) were pasted on the aluminum foil current collector, dried overnight in a vacuum oven at 80 °C, and punched and pressed uniaxially.

The operando coin cell was assembled using as-prepared $LiNi_{0.5}Mn_{1.5}O_4$ as the positive electrode and Li metal (MTI, 99.9% purity, 0.6-mm thickness, 16-mm diameter) as the negative electrode (Fig. S2). The cell top and base were from the standard CR2032 cells, which had a diameter of 20 mm and a height of 3.2 mm. Both sides had a hole drilled at the center of size around 3 mm in diameter, which was sealed by Kapton tape. The hole on the positive electrode was slightly smaller for a better pressure control. The washers were made of stainless steel 304 with aluminum coating and the dimensions were 14.5 mm O.D and 10.25 mm I.D with 0.3 mm thickness. The separator (Celgard C480) contained electrolyte of 1 M solution of lithium hexafluorophosphate ($LiPF_6$) in a 1:1 volume mixture of ethylene carbonate (EC) and dimethyl carbonate (DMC) (Gotion, battery grade). The cell was placed so that the material of interest, the $LiNi_{0.5}Mn_{1.5}O_4$ nanoparticles, was located downstream from the incident X-rays.

### Bragg Coherent Diffractive Imaging experiment

The operando BCDI experiment was conducted at Sector 34-ID-C in the Advanced Photon Source at Argonne National Laboratory. A double crystal monochromator was used to select X-rays with energy of $E = 9$ keV. The coherent X-rays with a focus size of 800 nm were incident on a fully operational half-cell. The rocking curve around the 111 Bragg peak was collected by a 2D detector (Timepix, 256 × 256 pixels, each pixel 55 μm × 55 μm) around $2\theta = 17$ degrees ($\Delta\theta = \pm 0.3°$). The detector was placed 1.1 m away from the sample and an evacuated flight tube was inserted between the sample and the detector. A total of 76 diffraction patterns were collected for a single 3D rocking scan with 1 s exposure time for each image. The 3D diffraction pattern of the same particle was continuously captured under operando conditions, while the coin cell battery was discharging. Each scan had a duration of about 2 min. Two rounds of alignment scans in the labx, labz (sample position relative to the incident beam) and theta directions were taken between every three rocking scans to ensure that the particle did not move out of the beam or rotate away from the diffraction condition. Low discharge rates were chosen to ensure that the particle remained around the same discharge state throughout one scan. Two discharge rates, C/10 (18 mA/g) and C/2 (80 mA/g), were chosen to illustrate the

two-phase behavior under different rates. A C/10 rate indicates that the battery finishes discharge in 10 h and a C/2 rate indicates that the battery finishes discharge in 2 h. We used a constant current (40 mA/g) to charge the cell to 5 V followed by 6-h holding period at constant voltage then discharged at constant current to 3.5 V followed by 75-min holding period at constant voltage for C/10 (18 mA/g) and 40-min holding period at constant voltage for C/2 (80 mA/g). All measurements were conducted under ambient conditions.

## Phase retrieval

The details of the correlated data inversion algorithm were reported elsewhere[14]. Here, for brevity we summarize the main aspects of the algorithm we used to invert the operando data. 10 diffraction scans that describe the entire two-phase reaction were inverted simultaneously. Every diffraction scan was aligned such that the center of each diffraction had the same scattering vector (during the experiment, the scattering angle was shifted to follow the peak evolution). The data inversion started with each scan running 30 reconstructions individually, each initiated with a random phase. For every 10 iterations, the error matrix of the correlation within the 30 reconstructions was calculated for each scan. Then we averaged the support of the 5 best-correlated reconstructions in each scan across all 10 scans (in total 50 reconstructions) while leaving the displacement fields unchanged. Then the averaged support was multiplied by the individual support and became the input support for the next set of reconstructions. The reconstruction consisted of a total of 610 iterations with alternating 10 iterations of the ER algorithm and 50 iterations of the RAAR algorithm. For the primary particle in the main text, five datasets that contain mutually different scans were reconstructed and then stitched back together into a single sequence. The series shown in Figs. 2 and S9 is a collage of multiple independent reconstruction procedures, i.e., some have no overlap of the diffraction data taken for the reconstructions. The continuous evolution of the nucleation and growth is a testament for the robustness of the algorithm. For the supplementary particle, one dataset that consists of 13 scans and one dataset that consists of 15 scans were reconstructed. The algorithm was performed 10 times on each dataset and the final imaging is the result of $5 \times 10$ reconstructions of each scan.

## Data availability

The diffraction and reconstruction data generated in this study have been deposited in the Open Science Framework database under accession code https://osf.io/a7knf/.

## Code availability

The reconstruction method is described in ref. 14. Specific MATLAB code can be requested from the corresponding author.

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

## Acknowledgements

The work at Cornell was supported by the National Science Foun-dation under award number CAREER DMR-1944907. A.S. acknowl-edges support from the Sloan Research Fellowship. The work at UC San Diego was supported by the Sustainable Power and Energy Center (SPEC) and US Department of Energy, Office of Science, Office of Basic Energy Sciences, under contract No. DE-SC0001805. This research used resources of the Advanced Photon Source, a US Department of Energy (DOE) Office of Science user facility operated for the DOE Office of Science by Argonne National Laboratory under Contract No. DE-AC02-06CH11357.

## Author contributions

A.S., Y.S.M., and O.S. conceived of the idea. S.H. and Y.S.M. prepared the samples. S.H., N.H., J.W., R.H., and A.S. performed the synchrotron-based measurements. Y.S. analyzed the data and wrote the paper. Y.S., S.H., N.H., J.W., R.H., Y.S.M., O.S., and A.S. participated in the inter-pretation of the data and revised the manuscript.

## Competing interests

The authors declare no competing interests.
