## [Peer review file · Nature Communications]

Operando real-space imaging of a structural phase transformation in the high-voltage electrode $\text{Li}_x\text{Ni}_{0.5}\text{Mn}_{1.5}\text{O}_4$

Corresponding Author: Professor Andrej Singer

Version 0:

Reviewer comments:

Reviewer #3

(Remarks to the Author)

This study reports on operando Bragg coherent diffractive imaging (BCDI) of a two-phase reaction during electrochemical Li-insertion in a high-voltage $\text{LiNi}_{0.5}\text{Mn}_{1.5}\text{O}_4$ cathode. The operando BCDI technique enables visualization of the nucleation and growth of the Li-rich phase within a 500-nm single-crystal particle at the nanometer scale. The principal finding of this study is that dislocation kinetics influence the shape of the reaction front, which is the boundary between the Li-rich and Li-poor phases. This phenomenon is well explained by the microelasticity model.

In their response letter, the authors have well addressed the concerns raised by Reviewer #1. I recommend that this paper be published in Nature Communications after a few points have been clarified.

1. In connection with Reviewer #1's comment, it would be helpful if the quantitative values on the horizontal axis were provided in Fig. 1.
2. The rebuttal provides an explanation for the appearance and subsequent disappearance of the second peak-like intensity at DoD 0–35% (Figure S4), but this explanation is not included in the manuscript. It should be incorporated into the manuscript.
3. It is very interesting that, despite the cathode particle with cubic crystal structure being in contact with the liquid electrolyte on most surfaces, the reaction proceeds from a single point rather than forming a core-shell structure. The reason of this observation should be mentioned in the manuscript. I believe this result may be related to the rate relationship between nucleation and phase growth.
4. Why do the q values of the Li-rich phase in Fig. 1 and Fig. S11 not match? For example, in Fig. 1, the q value is approximately 1.34 \AA^{-1} for Li-rich phase, while in Fig. S11, it appears to be around 1.35 \AA^{-1} . The same discrepancy is observed in Fig. S5. Could you clarify this difference?
5. In BCDI, the two-phase reaction is observed at a DoD greater than 37.9% (Fig. S4). However, in the diffraction data shown in Fig. S5, the two-phase reaction appears to develop much earlier, at approximately 10 mAh/g. This discrepancy might suggest that the reactions are not uniform within the coin cell.
6. In Fig. S8, the Fourier transform of the reconstructed image shows stronger interference fringe intensity. What is the cause of this increased intensity?
7. An important consideration for materials undergoing a two-phase reaction is whether an intermediate layer exists between the Li-rich and Li-poor phases. Given the spatial resolution of BCDI, the manuscript should explicitly state whether an intermediate layer is present, absent, or undetermined based on this analysis.
8. The lattice constants should be included in the color bar for the strain shown in Figs. 2 and 4.

9. The high distortion region that appears in the upper slice at 31.6% DoD in Fig. S12 disappears at 35.1%. Is this disappearance a physically meaningful phenomenon?

Version 1:

Reviewer comments:

Reviewer #3

(Remarks to the Author)

The authors have addressed all my concerns. I have no more comments. I recommend that this paper be published in Nature Communications.

Please, see the point-by-point response below. Reviewer comments in blue, our response in black, and modifications to the manuscript and the supplementary information in red.

Reviewer Comments:

Reviewer #3 (Remarks to the Author):

This study reports on operando Bragg coherent diffractive imaging (BCDI) of a two-phase reaction during electrochemical Li-insertion in a high-voltage $\text{LiNi}_{0.5}\text{Mn}_{1.5}\text{O}_4$ cathode. The operando BCDI technique enables visualization of the nucleation and growth of the Li-rich phase within a 500-nm single-crystal particle at the nanometer scale. The principal finding of this study is that dislocation kinetics influence the shape of the reaction front, which is the boundary between the Li-rich and Li-poor phases. This phenomenon is well explained by the microelasticity model.

In their response letter, the authors have well addressed the concerns raised by Reviewer #1. I recommend that this paper be published in Nature Communications after a few points have been clarified.

We thank the reviewer for recommending our work for publication in *Nature Communications*. We have addressed the points raised by the reviewer in the following point-by-point response. Additionally, we have revised our manuscript and supplementary information to comply with the editorial requests and *Nature Communications* formatting guidelines.

1. In connection with Reviewer #1's comment, it would be helpful if the quantitative values on the horizontal axis were provided in Fig. 1.

Author response: We agree with the reviewers and have added the horizontal axis in Fig. 1 and in Fig. S4.

2. The rebuttal provides an explanation for the appearance and subsequent disappearance of the second peak-like intensity at DoD 0–35% (Figure S4), but this explanation is not included in the manuscript. It should be incorporated into the manuscript.

Author response: We thank the reviewer for pointing out the lack of explanation for the appearance and subsequent disappearance of the flare at DoD 0–35% in the manuscript. We have included it in the manuscript:

...the main peak shows variations in its interference fringes suggesting local structural changes within the nanoparticle due to lithium intercalation (Fig. 1d and fig. S4). From 0% to 15.3% DoD (Fig. S4), a flare appears on the main peak. We interpret this flare as x-ray interference due to changes in the displacement field from Li intercalation, rather than the emergence of a second structural phase. There is a slight decrease of the peak position...

3. It is very interesting that, despite the cathode particle with cubic crystal structure being in contact with the liquid electrolyte on most surfaces, the reaction proceeds from a single point rather than forming a core-shell structure. The reason of this observation should be mentioned in the manuscript. I believe this result may be related to the rate relationship between nucleation and phase growth.

Author response: We agree with the reviewer that the observation of the reaction proceeding from a single point is intriguing and related to nucleation and growth kinetics. We have expanded the discussion of the rate relationship between nucleation and phase growth in the manuscript:

...Under this regime, models like the “domino-cascade” can explain the transformation of anisotropic ionic diffusion materials^{39,40}. For materials such as $\text{Li}_x\text{Ni}_{0.5}\text{Mn}_{1.5}\text{O}_4$ with a cubic crystal structure, ionic diffusion is typically isotropic. Nevertheless, our *operando* imaging shows that the reaction nucleates from a single localized point and grows through interface propagation⁴¹, even though most of the particle’s surface is in contact with the liquid electrolyte. At (dis)charge rates $C/2$ and $C/10$ we investigated, the transformation occurs in the charge transfer limited regime, where the ion diffusion within the particle is faster than its transfer from the electrolyte to the particle. In this case, the gradient in the chemical potential is small, and nucleation is rare due to the small driving force, making the reaction nucleation-limited. This contrasts with the diffusion-limited regime, where a larger driving force leads to many nucleation sites, but their growth is constrained. During the phase transformation, we directly observe...

4. Why do the q values of the Li-rich phase in Fig. 1 and Fig. S11 not match? For example, in Fig. 1, the q value is approximately 1.34 \AA^{-1} for Li-rich phase, while in Fig. S11, it appears to be around 1.35 \AA^{-1} . The same discrepancy is observed in Fig. S5. Could you clarify this difference?

Author response: We thank the reviewer for pointing out the discrepancy in the q values for the Li-rich phase between Fig. 1 and Fig. S11. The diffraction data in Fig. 1 and Fig. S5 is taken on a different nanoparticle from the diffraction data shown in Fig. S11. As shown in ref 4, different nanoparticles within a cell can have slightly different phase transformation behavior. In both particles, the fully changed phase (Li-poor phase) has similar q values of around 1.362 \AA^{-1} , but as the reaction proceeds, the Li-rich phase shows a slight variation in the q values (Fig. S5 and Fig. S11). We have modified the q vector axis and included the clarification in the supplement:

... the voltage plateau in the electrochemical data. Compared to the primary particle investigated in the main text and Fig. S5, the Li-rich phase of the supplementary particle displays a slightly different q value. This difference arises because particles within the cells can undergo subtly different lattice dynamics during the phase transformation⁴.

5. In BCDI, the two-phase reaction is observed at a DoD greater than 37.9% (Fig. S4). However, in the diffraction data shown in Fig. S5, the two-phase reaction appears to develop much earlier,

at approximately 10 mAh/g. This discrepancy might suggest that the reactions are not uniform within the coin cell.

Author response: The reviewer is right suggesting the reaction is not uniform within the cell. As described in the response to the previous question, different particles can display slightly different dynamics during the phase transformation as reported previously⁴ and now mentioned in the supplement of this work. In addition to the inhomogeneity within the cell, fig. S5 shows the projection while fig. S4 shows single slices (we found that the interference fringes best show the changes in the displacement field). Therefore, the single slice fig. S4 may not pass through both phases at the early stage of the reaction. We have included additional supplementary figures in fig. S4 that shows the two-dimensional average. The two-dimensional average exhibits intensity from the Li-rich phase as early as 11.2% DoD, which corresponds to the appearance of intensity around 13mAh/g in fig. S5. We have also clarified in the supplementary:

...particle within the cell (bottom). The diffraction is calculated by collapsing the data in Fig. S4 into one dimension and shows good agreement with Fig. S4, where the 3D diffraction data is projected in 2D. The voltage plateau...

6. In Fig. S8, the Fourier transform of the reconstructed image shows stronger interference fringe intensity. What is the cause of this increased intensity?

Author response: The stronger interference fringe intensity in the reconstruction originates from that the computed shape has a sharper boundary than the shape of the real particle. We have included the clarification in the supplementary information:

... the diffraction data is indicative of successful phase retrieval. The false colors are identical to Fig. S4. **Note that the diffraction pattern calculated from the reconstructed model shows a stronger interference fringe intensity than the measured diffraction data. This is because the computed shape has sharper boundaries compared to the shape of the real particle.**

7. An important consideration for materials undergoing a two-phase reaction is whether an intermediate layer exists between the Li-rich and Li-poor phases. Given the spatial resolution of BCDI, the manuscript should explicitly state whether an intermediate layer is present, absent, or undetermined based on this analysis.

Author response: We thank the reviewer for bringing the discussion on BCDI spatial resolution on the intermediate layer during a two-phase reaction. Our *operando* imaging shows that the interface width is less than 100 nm, around the spatial resolution of *operando* BCDI. Based on this, we consider no intermediate layer between the Li-rich and Li-poor phases under current BCDI limits, hence the material undergoes a two-phase reaction. Furthermore, if there were an intermediate layer characterized by a distinct lattice parameter, it should generate an additional peak in the diffraction, which we did not observe in our data. It is important to note that BCDI spatial resolution is coupled with the diffraction signal, i.e., higher diffraction intensity yields higher spatial resolution. We expect that the future development of the synchrotron sources will

generate a larger diffraction signal that allows us to study a sharper intermediate layer. We have included this discussion in the manuscript:

... Moreover, the recently developed diffraction-limited synchrotron sources will boost BCDI's time resolution to seconds⁴², enhancing our ability to observe these transformations at faster rates. **Finally, an increased diffraction signal will yield higher spatial resolution and enable the detection of diffraction from potentially present interphases during the phase transformation; at our current resolution, we observed no such phases.** Our results, therefore, ...

8. The lattice constants should be included in the color bar for the strain shown in Figs. 2 and 4.

Author response: We thank the reviewer for the suggestion to include lattice constants. We have included them in the color bar in Figs. 2 and 4, as well as in the figure caption:

... illustrated on three chosen planes. We show the strain field, $\varepsilon_{111}(\mathbf{r})$, as the local lattice constant compared to the average lattice constant, d_a , of the (111) planes between the fully charged, Li-poor phase ($d_c = 8.00 \text{ \AA}$) and the partially discharged Li-rich phase ($d_d = 8.10 \text{ \AA}$), where $d_a = (d_c + d_d)/2 = 8.05 \text{ \AA}$ and $\varepsilon_{111}(r) = d(r)/d_a - 1$. At 0% DoD...

9. The high distortion region that appears in the upper slice at 31.6% DoD in Fig. S12 disappears at 35.1%. Is this disappearance a physically meaningful phenomenon?

Author response: The nucleus appears to reshape and migrate during the phase transformation, which could be due to the minimization of the overall surface tension. We have observed similar behavior where two nuclei move and merge during phase transformation for the particle investigated in the main text. We have included the discussion in the supplement:

... particle also exhibits a nucleation and growth regime, where the nucleated Li-rich phase grows through interface propagation. **Between 26.8% DoD to 58.1% DoD, the nucleus moves and reorganizes, likely driven by surface tension reduction.** The images appear less smooth due to...